# Investigation of the Properties of Multilayer Nanostructured Coating Based on the (Ti,Y,Al)N System with High Content of Yttrium

Sergey Grigoriev [1] , Alexey Vereschaka [2,*] , Filipp Milovich [3], Nikolay Sitnikov [4], Jury Bublikov [2], Anton Seleznev [1] , Catherine Sotova [1] and Alexander Rykunov [5]

[1] Department of High-Efficiency Processing Technologies (VTO), Moscow State University of Technology, STANKIN, Vadkovsky per. 1, 127055 Moscow, Russia
[2] Institute of Design and Technological Informatics, Russian Academy of Sciences (IDTI RAS), 127055 Moscow, Russia
[3] Materials Science and Metallurgy Shared Use Research and Development Center, National University of Science and Technology MISiS, Leninsky Prospect 4, 119049 Moscow, Russia
[4] State Scientific Center, Russian Federation "Keldysh Research Center", 8, Onezhskaya Str., 125438 Moscow, Russia
[5] Federal State-Financed Educational Institution, High Professional Education P. A. Solovyov Rybinsk State Aviation Technical University, St. Pushkina, 53, 152934 Rybinsk, Russia
* Correspondence: dr.a.veres@yandex.ru

**Abstract:** The studies are focused on the properties of the multilayer composite coating based on the (Ti,Y,Al)N system with high content of yttrium (about 40 at.%) of yttrium (Y). The hardness and elastic modulus were defined, and the resistance to fracture was studied during the scratch testing. Two cubic solid solutions (fcc phases), including c-(Ti,Y,Al)N and c-(Y,Ti,Al)N, are formed in the coating. The investigation of the wear resistance of the (Ti,Y,Al)N-coated tools during the turning of steel in comparison with the wear resistance of the tools with the based on the (Ti,Cr,Al)N system coating and the uncoated tools found a noticeable increase (by 250%–270%) in rake wear resistance. Active oxidation processes are observed in the (Ti,Y,Al)N coating during wear. It can be assumed that yttrium oxide is predominantly formed with a possible insignificant formation of titanium and aluminum oxides. At the same time, complete oxidation of c-(Y,Ti,Al)N nanolayers is not observed. Some hypotheses explaining the rather high performance of a coating with a high yttrium content are considered.

**Keywords:** nanostructured coatings; yttrium nitride; tool wear; oxidation; metal cutting

## 1. Introduction

The increasing requirements for tool materials in general and for coatings in particular predetermine the need to find ways for further improvement in their properties [1–5]. Coatings with a nanolayer structure, which combines nanolayers with different properties, are widely used. In particular, it has been found that plastic flow in a (Cr,Al)N nanolayer coating with alternating layers dominated by CrN and AlN is not associated with the movement of classical dislocations, but occurs as a result of rotation of nanosized grains and sliding along the boundary for larger grains [6]. Changing the parameters of the nanolayer structure of the (Ti,Cr,Al)N coating makes it possible to control its properties [7,8].

It is known that YN forms a cubic crystal fcc lattice with the parameter of a = 4.88 Å [9]. In the Y-containing coatings, yttrium can either form its own nitride phase of YN or enter into the composition of other phases (for example, as an element of a substitutional solid solution). For example, no single (Y,Al)N phase is detected in the system of (Y,Al)N, and AlN reacts with Y to form the phases of YN and $YAl_2$ [10,11]. When the coating, which includes the (Al,Y)N phase, is heated to 600–1000 °C upon contact with the atmosphere,

oxides $Al_2O_3$ and $Y_2O_3$ are formed on the surface [12]. The introduction of Y in the composition of nitride coating increases its resistance to oxidation. The temperature of the onset of active oxidation of the (Nb,Y)N coating, which was 500 °C for a system without the addition of yttrium, increases to 760 °C at a Y content of 12.1 at.% [13]. An increase in the yttrium content in the nitride coating leads to the formation of an amorphous microstructure with a finer and denser morphology and a smoother surface [14].

When Y is added to the (Ti,Cr,Al)N system, its heat resistance increased to 1200 °C [15]. Upon heating up to 1000 °C, the number of oxides formed in the Y-containing coating was half the size in comparison with the Y-free coating. When Y is added to the (Cr,Al)N system, the cohesive energy increases (the energy of formation decreases) due to the substitution of Cr atoms for Y atoms in the crystal lattice, and, as a result, the phase stability increases [16]. The described effects contribute to a noticeable growth of the wear resistance of the coating, especially when the coating is exposed to high temperatures [17].

The addition of Y to the coating composition leads to a decrease in the grain size [18]. The grain size decreases from 76 to 21 nm as the Y content increases from 0 to 5.8 at.%. Upon heating to the temperature of 1200 °C, the (Cr,Al)N coating demonstrates the growth of grains from the initial size of 30–40 nm to 100 nm. In the (Cr,Al,Y)N coating, the grain sizes remain at the level of 30–40 nm at the temperature of 1200 °C. The great hardness of the (Cr,Al,Y)N coating is retained at temperatures up to 1100 °C [13]. The addition of Y to the composition of nitride coatings also noticeably slows down the diffusion processes [13,19].

Thus, the introduction of yttrium into the composition of the coatings can significantly enhance the cutting properties of the tools due to the improvement in the tribological properties and resistance to oxidation and diffusion, which may be explained by the formation of complex oxide films in the cutting zone [10,15,20–25].

Several studies consider the properties of the (Ti,Al,Y)N coating. It has been found that the introduction of yttrium into the (Al,Ti)N coating leads to a decrease in the average grain size and compaction of the microstructure, as well as an increase in the oxidation resistance of the coating [26,27]. The introduction of yttrium also increases heat resistance and hardness at high temperatures (above 1000 °C) [28]. Yttrium slows down the decomposition of the $Ti_{1-x-y}Al_xY_yN$ solid solution and contributes to the formation in the coating of c-TiN, c-YN, and w-AlN phases. The introduction of Y reduces the compressive stress, increases the hardness, and also enhances the predominant formation of dense $Al_2O_3$ oxide films instead of porous $TiO_2$ oxide [12]. In general, the introduction of yttrium increases the resistance to oxidation [29,30]. It is found that the wear resistance of the (Ti,Al,Y)N coating grows with an increase in the Y content [31]. At the same time, the average rates of abrasive wear of the (Ti,Al,Y)N coating were 3–5 times lower in comparison with the (Ti,Al)N coating and 10 times lower in comparison with the TiN coating. In [31], such properties of the (Ti,Al,Y)N coating are associated with the formation of a dense structure with nanosized grains and a decrease in surface roughness and the level of internal stresses.

It should be noted that almost all of the above studies consider the coatings with the low (1–2 at%) content of yttrium. Such choice may be explained by the fact that yttrium nitride (YN) hydrolyzes upon contact with water and forms $Y_2O_3$ oxide upon heating in an oxygen-containing atmosphere [32–35]. The hardness of yttrium oxide is significantly lower in comparison with the hardness of YN or TiN (hardness of $Y_2O_3$ stays in a range from 3 to 8 GPa [36]). At the same time, $Y_2O_3$ is characterized by a very high thermal stability (up to 2100 °C) and rather low thermal conductivity, which makes it possible to effectively use it as a barrier to heat fluxes in the coating [37]. A possible useful property of YN and $Y_2O_3$ is their relatively high fracture toughness [38,39].

Of particular interest could be an investigation focused on the properties of a coating in which the layers with great hardness and wear resistance (for example, the layers based on the (Ti,Al)N system) would be alternated with the layers with good barrier properties combined with high fracture toughness (for example, the layers based on YN). It has been established that yttrium and yttrium nitride increase the fracture toughness when they are introduced into the composition of alloys and nitride phases [40–44].

Fracture toughness increases significantly (from 0.1 to 1.0 MPa.m$^{1/2}$) with an increase in the yttrium content from 0 to 20 at.% in the TiN-based coating [39].

The alternation of the mentioned nanolayers will protect YN-based layers from rapid oxidation, while allowing the formation of tribologically active $Y_2O_3$ oxides that have a positive effect on the conditions in the cutting zone [22–24].

It has been found that the introduction of up to 7.8 at.% Y into TiN contributes to the formation of the dominant fcc phase of the c-(Ti,Y)N solid solution, with the formation of a mixture of c-TiN and c-YN fcc phases upon an increase in the yttrium content [39,45]. The hardness increases from 21 GPa for the yttrium-free coating to 26 GPa for the coating with 10.2 at.% yttrium. The fracture toughness of the coating increases with a growth of the yttrium content.

An additional positive effect is provided by the application of a multilayer coating architecture, which includes an adhesive layer with a substrate, a transition layer, and a wear-resistant layer [46–50]. In this case, the wear-resistant layer has a nanolayer structure with alternating nanolayers characterized by different mechanical properties [50–52].

Therefore, the Ti-TiN-(Ti,Y,Al)N coating was chosen for the study. This coating has a three-layer architecture [46,53], including the Ti adhesive layer, the TiN transition layer, and the (Ti,Y,Al)N wear-resistant layer. The (Ti,Y,Al)N wear-resistant layer has a nanolayer structure with the modulation period λ of about 60 nm. Due to the yttrium content in the wear-resistant layer (about 40 at.%), the formation of two phases (c-(Ti,Al)N and c-YN) can be predicted.

The object of comparison was the Ti-TiN-(Ti,Cr,Al)N coating with similar architecture parameters. This composition was chosen due to the wide application of the coatings based on the (Ti,Cr,Al)N system in the manufacturing of metal-cutting tools [10,54–58].

For convenience, the coatings under comparison will be further designated only by their wear-resistant layer: (Ti,Y,Al)N and (Ti,Cr,Al)N, respectively.

## 2. Materials and Methods

### 2.1. Sample Preparation

The deposition of the coatings under comparison was carried out on the specialized PVD unit of VIT-2 [46,48–50,59,60] (IDTI RAS–MSTU STANKIN, Moscow, Russia), using the filtered cathodic vacuum arc deposition (FCVAD) system (IDTI RAS–MSTU STANKIN, Moscow, Russia) [48–50] for the aluminum cathode and the Controlled Accelerated Arc system (CAA-PVD) (IDTI RAS–MSTU STANKIN, Moscow, Russia) [61,62] –for other cathodes. A detailed circuit diagram and layout of the VIT-2 unit is presented in [46].

The following cathodes were used for coating deposition: Al (99.80%), Cr (99.90%), Ti (99.60%), and Y (99.98%). Due to the tendency of yttrium to hydrolyze at high temperatures, the yttrium cathode had a copper mounting part with a cooling system to prevent any contact of yttrium with water. Chemically pure nitrogen of high purity (grade 6.0) was used. Volume fraction of nitrogen, % not less than 99.99990.

The substrate used was carbide inserts SNUN ISO 1832:2012, 12.00 mm × 12.00 mm × 4.75 mm (WC + 15% TiC + 6% Co) (KZTS, Kirovograd, Russia). Each type of coating was deposited on 8 samples.

Before coating deposition, preparation procedures were carried out, including:

- Pre-washing when exposed to ultrasound in a special solution;
- Washing in purified water;
- Drying and wiping with pure medical alcohol;
- After that, the samples were placed on the special fixture and loaded into the chamber of the unit;
- Before coating deposition, the samples are finely cleaned and thermally activated in a gas and metal plasma flow, with the following process parameters: gas (Ar) pressure = 2.0 Pa, cathode arc current = 110 A, voltage on substrate U = 110 V;
- After the above stages of preparation, coatings were deposited on the samples;
- Key characteristics of the coating process are presented in Table 1.

**Table 1.** Main parameters of the coating deposition process.

| Nitrogen Pressure (Pa) | Voltage on Substrate $U$ (V) | Cathode Arc Current (A) | | | |
|---|---|---|---|---|---|
| | | Al | Ti | Cr | Y |
| 0.42 | −150 DC | 160 | 110 | 75 | 85 |

The rotation speed of the turntable was 0.7 rpm [7,48].

The surface temperature of the substrate during the deposition of the coating was 650–700 °C.

In the deposition of adhesive and transition layers, only the Ti cathode was used; deposition took place in an argon and nitrogen atmosphere, respectively. When depositing (Ti,Y,Al)N wear-resistant layer, three cathodes were used: Ti, Y, and Al.

*2.2. Characterization*

After coating deposition, characteristics such as hardness, modulus of elasticity, and fracture strength of the coatings during the scratch testing are investigated.

A CB-500 tester (Nanovea, Irvine, CA, USA) with a nanomodulus was used to measure the elastic modulus and hardness of the samples. A precision piezoelectric drive and a high sensitivity load cell were used. Instrumental indentation with a Berkovich indenter was used at a load of 50 mN. Since the study of the hardness of the coating as a thin object presents a certain difficulty, 20 measurements were carried out for each sample, then the average values were determined and the deviations of the values were indicated.

The scratch resistance was determined on a Nanovea instrument (Nanovea, Irvine, CA, USA) according to the ASTM C1624-05 method [63]. The indenter was a Rockwell C Diamond with a tip radius R = 100 μm and a taper angle of 120 degrees. The tests were carried out with a linearly increasing load from 0.2 to 40 N at a loading rate of 5 N/min. After microfurrow formation, dimensions were measured using an optical SX45 stereo microscope (Vision Engineering Ltd., Woking, UK). The moment of complete destruction of the coating corresponds to the load $L_{c2}$.

Then the structural characteristics and composition (elemental and phase) of the coatings were studied. To study the nanostructure of the coatings, a transmission electron microscope (TEM) JEM 2100 (JEOL, Tokyo, Japan) at an accelerating voltage of 200 kV was used. The elemental composition was determined with the EDX (energy-dispersive X-ray) system of INCAEnergy (OXFORD Instruments, Oxford, UK). Samples (lamellas) were cut out using Strata focused ion beam (FIB) 205 (FEI, Hillsboro, OR, USA).

A scanning electron microscope (SEM) Quanta 600 FEG (FEI, Hillsboro, OR, USA) was also used. Preparation of samples for metallographic studies was carried out on equipment for the production of metallographic sections Isomet 1000 (Bühler AG, Uzwil, Switzerland), automatic hydraulic press for hot fitting Simplimet 1000 (Bühler AG, Uzwil, Switzerland), automatic grinding and polishing machine EcoMet 250 and AutoMet 250 (Bühler AG, Uzwil, Switzerland).

Then, the wear resistance of the coated cutting tool during turning is investigated. The wear resistance of the samples under consideration was studied during the turning of Inconel 718 at a CU 500 MRD lathe (Sliven, Bulgaria) with a ZMM CU 500 MRD variable-speed drive, without cutting fluid (dry cutting), with the cutting parameters as follows: γ = –7°, α = 7°, λ = 0, r = 0.4 mm; at cutting mode: f = 0.25 mm/rev, $a_p$ = 1.0 mm, and $v_c$ = 300 m/min. The flank wear $VB_{max}$ = 0.3 mm was assumed as the criterion for the limit wear of the tools. Each test was repeated 5 times, and error bars are depicted in the graph.

To consider the wear process on the (Ti,Y,Al)N coating, a transverse section was made passing through the cutting edge, in parallel to the minor flank face (Figure 1). A flank wear land can be noticed on the transverse section, and there is no wear crater on the rake face.

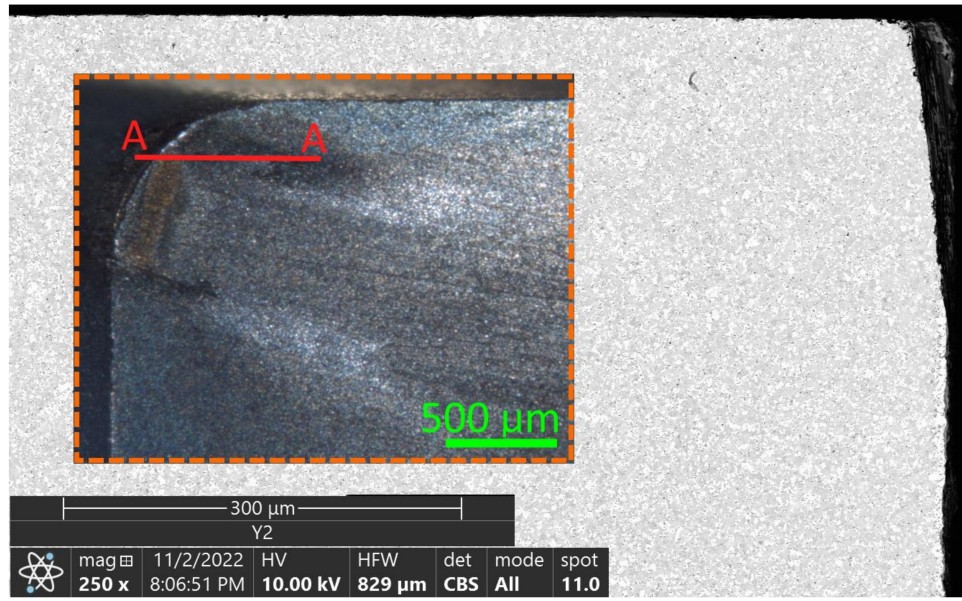

**Figure 1.** Transverse section of a worn insert (SEM); (center) view of a worn insert coated with (Ti,Y,Al)N and the designation of the A-A transverse section plane.

## 3. Results and Discussion

At the elastic modulus of 356 ± 24 GPa, the hardness of the (Ti,Y,Al)N coating is considerably high (HV 2758 ± 78). The reference coating of (Ti,Cr,Al)N has the hardness of HV 3182 ± 53 and the elastic modulus of 438 ± 32 GPa. The compositions of the (Ti,Y,Al)N coating and the (Ti,Cr,Al)N reference coating are exhibited in Table 2. It can be noted that the compositions of the coatings under comparison in terms of the Ti and Al content are quite close (taking into account the gradient distribution of elements over the coating thickness). The study of the fracture strength of the coatings during the scratch testing reveals that both coatings have good adhesion to the substrate and that the point of complete coatings failure of $L_{C2}$ was not reached at the load of 50 N.

**Table 2.** Compositions of the coatings under comparison.

| Coating | Element Content, at.% | | | |
|---|---|---|---|---|
| | **Ti** | **Cr** | **Al** | **Y** |
| (Ti,Cr,Al)N | 63.75 | 23.98 | 12.27 | - |
| (Ti,Y,Al)N | 51.82 | - | 7.69 | 40.49 |

Figure 2a,b depict the nanolayer structure of the coatings under comparison. The (Ti,Y,Al)N coating exhibits noticeable differences in the structural features of its nanolayers, while the structure of the (Ti,Cr,Al)N coating is much more homogeneous. The value of the modulation period λ is 65 nm for the (Ti,Y,Al)N coating and 53 nm for the (Ti,Cr,Al)N coating. These values are quite close and are in the range of values that provide the optimal properties of the coating [7,48]. In terms of the phase composition (Figure 2c,d), there is a noticeable difference between the coatings under comparison. While a single fcc phase of the c-(Ti,Cr,Al)N solid solution is formed in the (Ti,Cr,Al)N coating, then two cubic solid solutions (fcc phases of c-(Ti,Y,Al)N and c-(Y,Ti,Al)N)) are formed in the (Ti,Y,Al)N coating. This two-phase structure of the (Ti,Y,Al)N coating correlates to some extent with its significantly differentiated nanolayer structure.

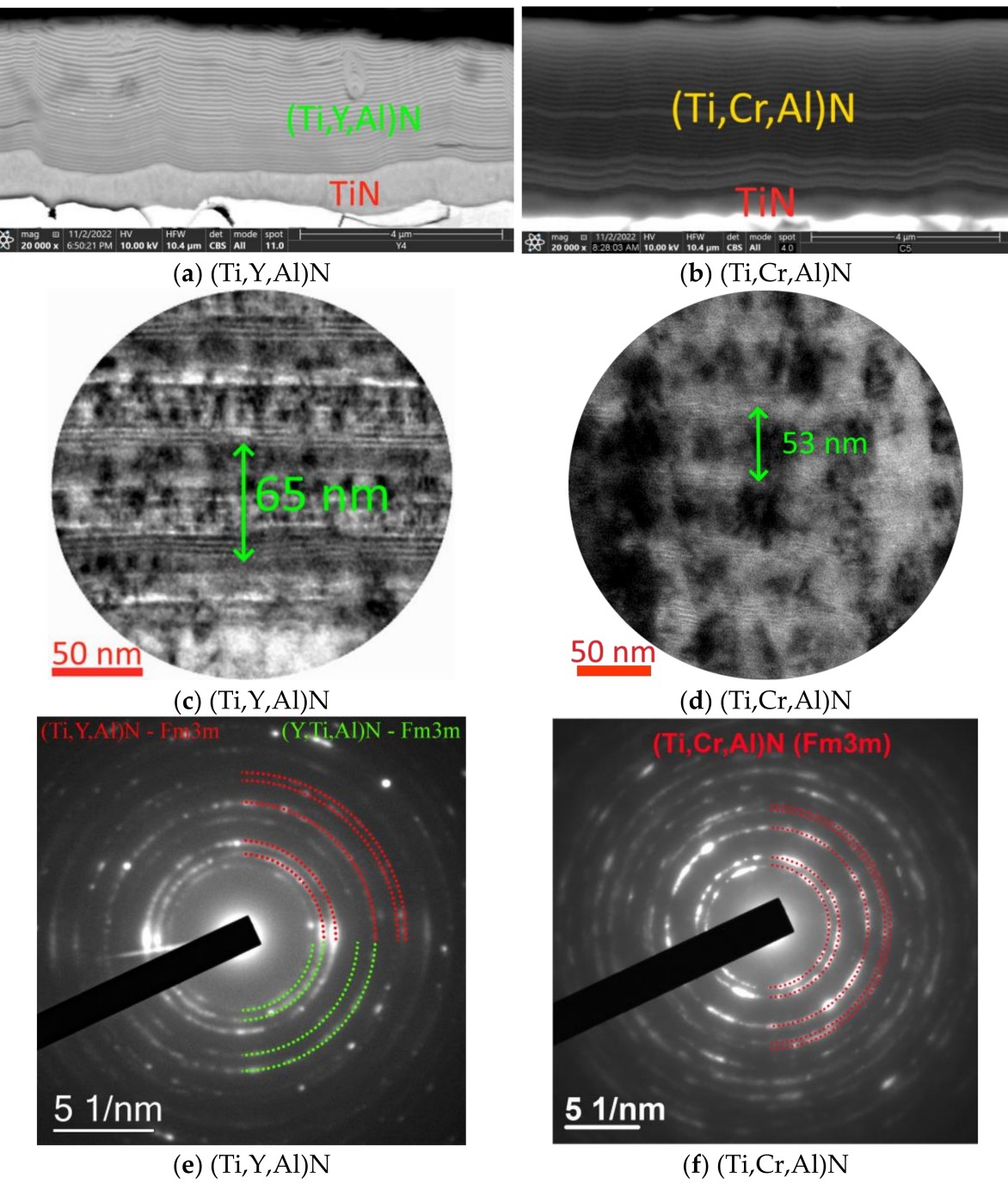

**Figure 2.** (**a**,**b**) View of transverse sections of the studied coatings (SEM), comparison of (**c**,**d**) the nanolayer structures (TEM) and (**e**,**f**) phase compositions, defined by the selected area electron diffraction pattern (SAED) method.

As a result of the cutting tests, the relationship between the tool flank wear VB and the cutting time was investigated (Figure 3). The tools with the coatings of (Ti,Y,Al)N and (Ti,Cr,Al)N, as well as uncoated tools were compared. The coated tools demonstrated significantly higher wear resistance in comparison with the uncoated tools. While the wear rates of the coated tools were fairly close, the tool with the reference coating of (Ti,Cr,Al)N demonstrated slightly higher wear resistance.

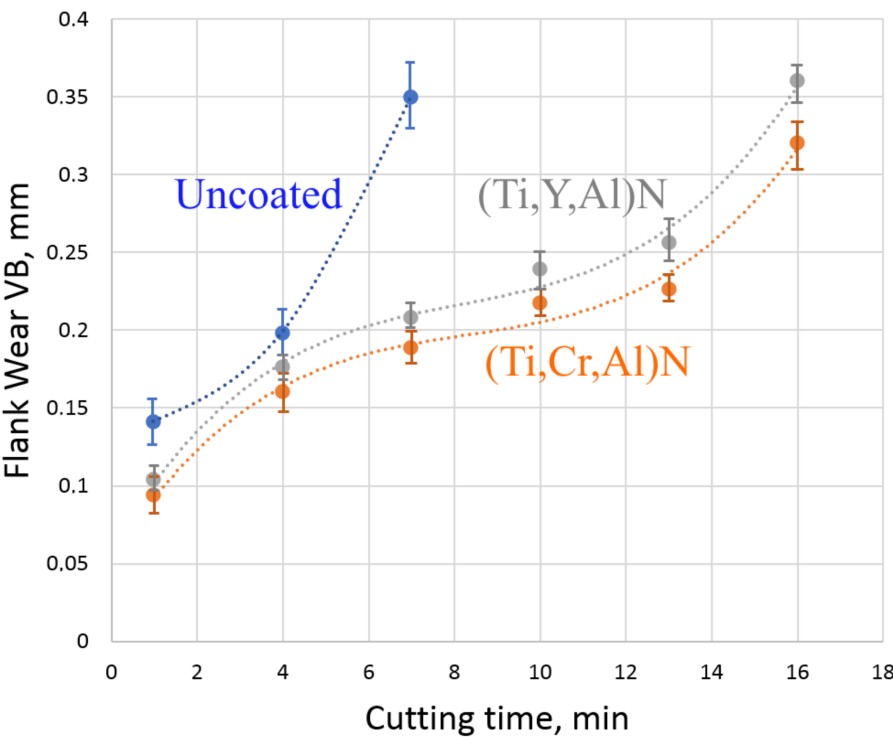

**Figure 3.** Wear dynamics along the tool flank faces of the tools with the coatings under comparison and the uncoated tools (after 16 min of cutting at f = 0.25 mm/rev, $a_P$ = 1.0 mm, and $v_c$ = 300 m/min).

The studies of the wear pattern of the coating conducted with a scanning electron microscope (SEM) (Figure 4) reveal the retained TiN transition layer in the worn area on the rake face. The (Ti,Y,Al)N wear-resistant layer has been preserved in fragments (see the area lighter in contrast). No delamination of the coating from the substrate is observed, which indicates good adhesion between the coating and the substrate. This confirms the data obtained earlier as a result of the scratch testing.

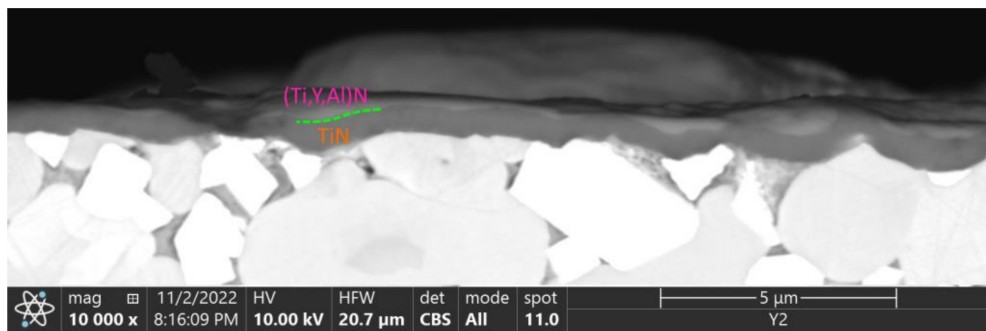

**Figure 4.** General wear pattern of the coating (SEM, after 16 min of cutting at f = 0.25 mm/rev, $a_p$ = 1.0 mm, and $v_c$ = 300 m/min).

The studies conducted with a transmission electron microscope (TEM) provide more information about the wear pattern on the coating. Figure 5 exhibits a general view of a lamella cut out from the area of the wear boundary of the coating. Areas A, B, and C are highlighted for further research. The TEM image correlates well with the previously obtained SEM. The well-preserved TiN transition layer can be clearly seen, with the structure, coarse-grained relative to the grains of the overlying (Ti,Y,Al)N coating. There are also preserved fragments of the (Ti,Y,Al)N wear-resistant layer, which has a nanolayer tructure.

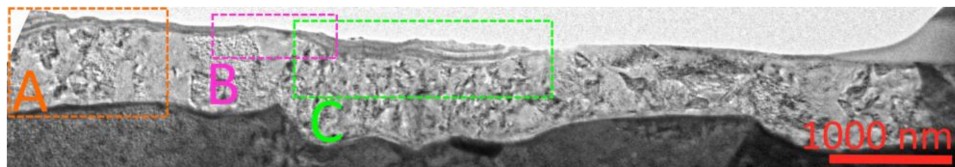

**Figure 5.** General view of the lamella and localization of Areas A, B, and C (after 16 min of cutting at f = 0.25 mm/rev, $a_p$ = 1.0 mm, and $v_c$ = 300 m/min) (TEM).

The examination of Areas A, B, and C in Figure 6 reveals a significant differentiation between the separate nanolayers of the coating. This differentiation is slightly enhanced compared to the coating after deposition (Figure 1). The described effect may be associated with the formation of yttrium oxide ($Y_2O_3$), which partially or completely replaces yttrium nitride (YN). The difference in the structure of the nanolayers is particularly noticeable in Image C2 (Figure 6d). The c-(Ti,Y,Al)N layer (darker in contrast) has the layered structure, while the (Y,Ti,Al)N layer (lighter in contrast) does not have layered structure in the region under consideration. The deeper layers of (Y,Ti,Al)N have the layered structure (Figure 6a,b), which may indicate the oxidation of the (Y,Ti,Al)N layers coming to the surface (and, accordingly, contacting oxygen in air), with the corresponding transition of YN → $Y_2O_3$.

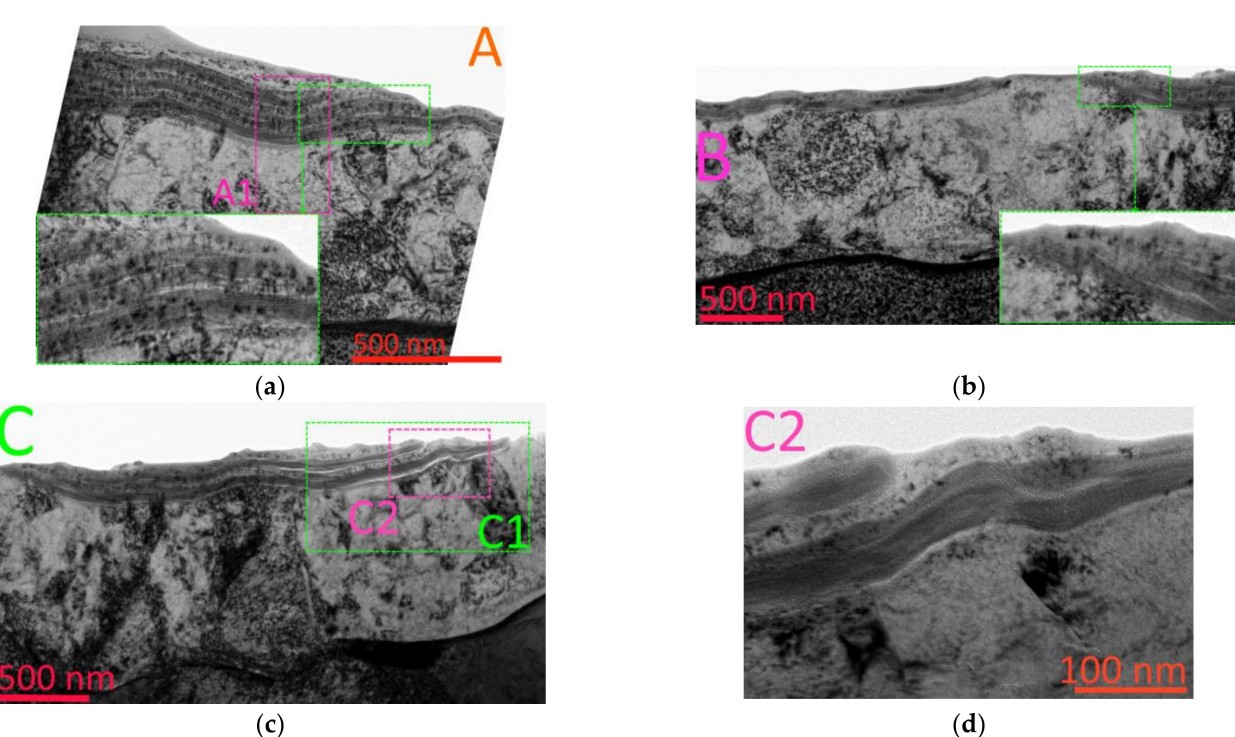

**Figure 6.** Investigation of the failure of the nanolayer structure of the coating. (**a,b**) (Y,Ti,Al)N nanolayers retain their layered structure, (**c,d**) (Y,Ti,Al)N nanolayers have lost their layered structure (possible YN → $Y_2O_3$ transformation) (TEM).

Due to the small area of the studied regions, no reliable SAED analysis can be conducted, but the analysis of the elemental composition makes it possible to identify zones of possible oxidation (Figure 7). The surface of the coating includes a layer up to 75 nm thick, in which an insignificant content of Y is combined with the high content of Ti and Al (see Point 1, Area A1 in Figure 7a). The content of oxygen in this layer is also high. It can be assumed that the oxides of $Al_2O_3$ and $TiO_2$ are formed as a result of oxidation and spinodal decomposition [64–69]. The next two Points 2 and 3 exhibit an increased content of yttrium with the high content of oxygen retained and a decrease in the content

of Ti and Al. It can be assumed that yttrium oxide ($Y_2O_3$) is predominantly formed in this region. The insignificant presence of iron and chromium in the outer layers of the coating can be explained by the diffusion of these elements from the material being machined. The analysis of the distribution of elements in the outer layers of the worn coating (Area C1 in Figure 7b) finds some increase in oxygen content at Point 2, in which an increased content of yttrium is also detected. Taking into account that Point 2 is located farther from the coating surface compared to Point 1, and the oxygen content at Point 2 is higher, it can be assumed that the high oxygen content is associated with the active transformation of YN $\rightarrow$ $Y_2O_3$. This conclusion is also confirmed by the results of the analysis of Regions 6 and 7. Whereas Region 6 (with the dominant yttrium content of 36.03 at.%) has a high oxygen concentration (45.63 at.%), then in Region 7 with the dominance of titanium (49.14 at.%) the content of oxygen is noticeably lower (26.20 at.%). This may also indicate the predominant formation of yttrium oxide with a significantly less intense formation of titanium and aluminum oxides, or the absence of these oxides in the considered region.

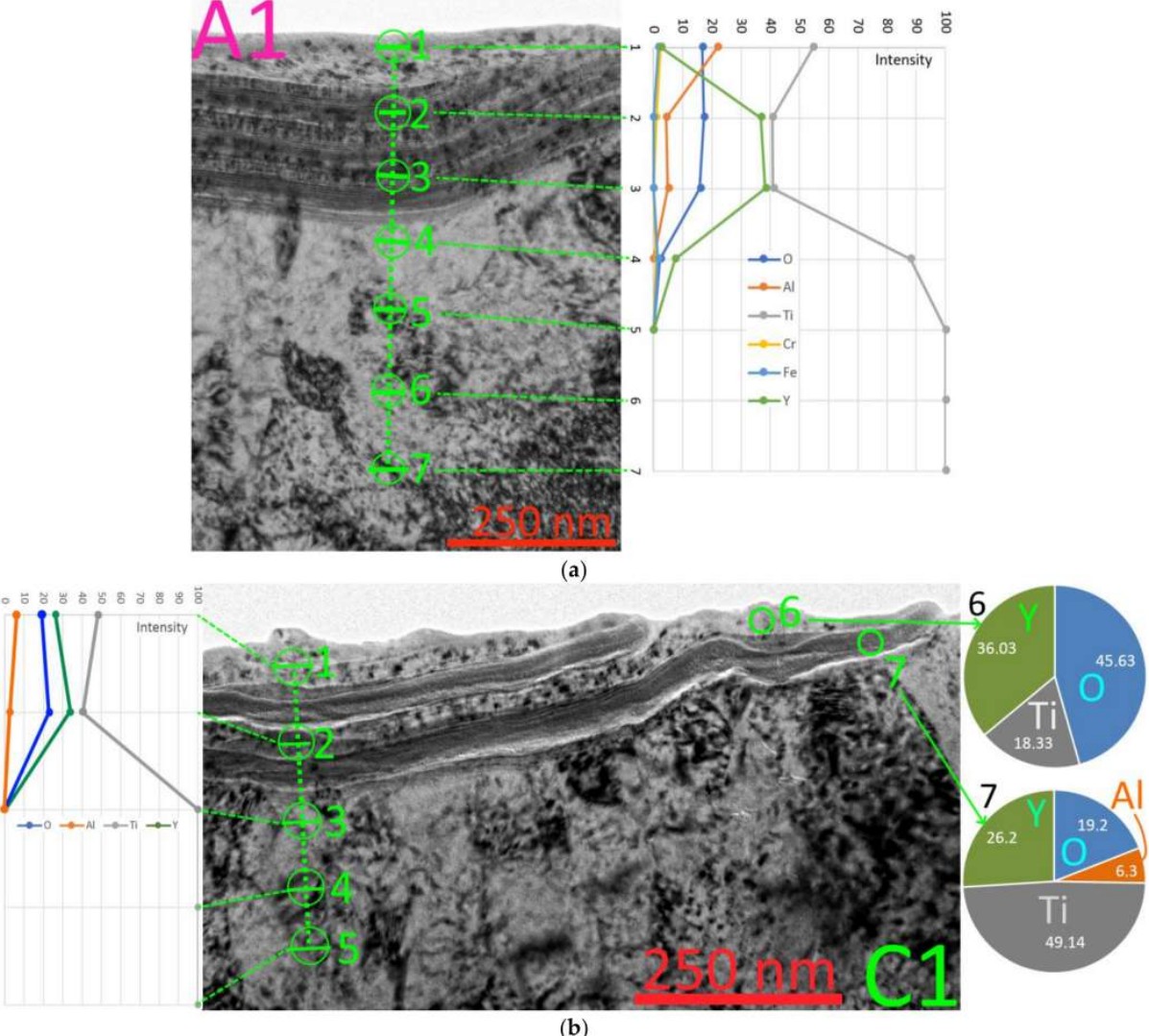

**Figure 7.** Analysis of the distribution of elements in the outer layers of the worn coating. (**a**) Area A1, (**b**) Area C1. The localization of Areas A1 and C1 is exhibited in Figure 6 (TEM).

Despite the active oxidation processes in the surface layers, the (Ti,Y,Al)N coating with a high yttrium content showed a rather high efficiency. As a direction for further research, it would be interesting to study the oxidation processes in the coating at different yttrium

contents. Thus, it is possible to establish both the yttrium content, which is optimal for the tool life, and the ratio of the negative (coating wear) and positive (tribological conditions optimization) influence of oxidative processes on the general properties of the coating.

## 4. Conclusions

The studies were focused on the properties of the Ti-TiN-(Ti,Y,Al)N multilayer composite coating with the high content (about 40 at.%) of yttrium in its wear-resistant layer.

- The Ti-TiN-(Ti,Y,Al)N coating is characterized by the considerably high hardness (HV 2758 $\pm$ 78) with the elastic modulus of 356 $\pm$ 24 GPa;
- Two cubic solid solutions (fcc phases)–c-(Ti,Y,Al)N and c-(Y,Ti,Al)N–are formed in the coating;
- The study of the wear resistance of the Ti-TiN-(Ti,Y,Al)N-coated tools during the turning of steel in comparison with the wear resistance of the tools with the reference coating of Ti-TiN-(Ti,Cr,Al)N and the uncoated tools detects a noticeable increase in the wear resistance on the rake face ( by 250%–270%) for the tools with both coatings. With the wear rates of the coated tools being fairly close, the tool life of the tool with the reference coating of (Ti,Cr,Al)N was slightly longer (by 10%–15%);
- During the process of wear, active oxidation processes take place in the layers of the Ti-TiN-(Ti,Y,Al)N coating that are in contact with the cut material flow. The mentioned processes consist in the dominant formation of yttrium oxide of $Y_2O_3$ with a possible slight formation of oxides of $Al_2O_3$ and $TiO_2$. Thus, for the described cutting conditions, the mechanisms of oxidative wear dominate in the coating.

Therefore, despite the tendency of YN to hydrolyze upon contact with water (including that contained in the atmosphere) and oxidize upon heating in an oxygen-containing environment, the Ti-TiN-(Ti,Y,Al)N coating demonstrated fairly good wear resistance. After 16 min of cutting the worn area on the rake face still demonstrates undamaged fragments of not only the transition layer, but also the wear-resistant layer of the coating. A possible reason for the lack of complete oxidation or noticeable hydrolysis of (Y,Ti,Al)N may be the protective functions of the layers with the dominance of (Ti,Al)N, which protect the underlying layers of (Y,Ti,Al)N from early oxidative damage. Another reason for the described phenomenon may be the substitution of the YN phase (which is prone to hydrolysis and thermal oxidation) for the solid solution phase of (Y,Ti,Al)N, the properties of which may be different. At the same time, the oxide layers formed on the surface of the coating improve the tribological conditions in the cutting zone and thus slow down the tool wear rate.

**Author Contributions:** Conceptualization, A.V.; methodology, A.V., F.M., N.S. and A.S.; investigation, F.M., J.B., N.S., C.S. and A.S.; resources, S.G.; data curation, C.S. and A.R.; writing—original draft preparation, A.V.; writing—review and editing, A.V.; project administration, S.G.; funding acquisition, S.G. All authors have read and agreed to the published version of the manuscript.

**Funding:** This work was supported financially by the Ministry of Science and Higher Education of the Russian Federation (project No FSFS-2021-0006).

**Institutional Review Board Statement:** Not applicable.

**Informed Consent Statement:** Not applicable.

**Data Availability Statement:** Not applicable.

**Acknowledgments:** The study used the equipment from the Centre for collective use of Moscow State Technological University STANKIN (agreement No. 075-15-2021-695, 26/07/2021). The coating structure was investigated using the equipment of the Centre for collective use of scientific equipment "Material Science and Metallurgy", purchased with the financial support of the Ministry of Science and Higher Education of the Russian Federation (GK 075-15-2021-696).

**Conflicts of Interest:** The authors declare no conflict of interest.

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
