# Peer review of "Investigation of the Properties of Multilayer Nanostructured Coating Based on the (Ti,Y,Al)N System with High Content of Yttrium"

_coatings, doi:10.3390/coatings13020335_

Round 1

Reviewer 1 Report

After reading this article carefully, I think a major revision is necessary before bringing this paper to the readers.

First, the title is unclear or cannot precisely cover the topics discussed in this paper. The majority of discussion is regarding the wear-resistant properties of the (Ti,Y,Al)N top layer in this multilayer structure and possible evidence of why this layer can provide such good properties. Please reconsider the name of this article. However, even if the appropriate name has been addressed here, this article still needs significant revision since the evidence to support the conclusion is insufficient, and some vital material characterizations are still missing.

1.Abstract:

I suggest the authors rewrite the abstract. The abstract should bring the motivation of the research and the highlighted part of the conclusion/discovery after revies the experiment part.

2. Introduction

I suggest the author simplify/rewrite the introduction session and give a more precise/more detailed picture of how it relates to your research motivation/topic. Too many articles have been mentioned; however, not all of them are related, and they are not well organized to fit with the scope of this paper.

Please add a clear explanation and background introduction of the Ti/N-based multilayer system in the beginning part of the instruction. The advantages of this multilayer system (address on high wear-resistance part) and how the Y, Cr play inside should be discussed next. The materials from rows 47-116 should be reorganized and simplified after introducing the system.

Please add the instruction to the deposition method used in this paper and related work.

3. Materials and Methods

The whole experiment setup and analysis are not straightforward for me. Please reorganize and separate this section into two subsections -sample preparation & characterization.

Please keep the model type number in parentheses for all the systems/tools used here.

Sample preparation part:

1. Please explain how the sample has been prepared step by step in time and what kind of sample has been prepared/used.

2. Please provide a few schematic drawings here to show the deposition system and sample preparation flow.

Sample characterization part:

1. Please explain which kinds of characterization are needed first and why they are needed, then give a description of the tools/settings. For example, the method of transverse section samples (rows 228-234) should be discussed in the sample preparation part.

2. Add material to illustrate how the hardness and cutting tests were conducted.

3. reorganize all the paragraphs.

4. Results and Discussion

Table 2: Please provide the detail of how you obtain the element content in each coating.

Figure 1: Please provide an XRD analysis here to support your conclusion.

Kindly suggest adding the Ti-TiN-( Ti, Y, Al)N with low Y content analysis as evidence to support your conclusion.

Figure 2: Please explain the criterion of why the data stopped recording at 7 and 16,16 mins, respectively.

Figure 3 needs to be reorganized and moved to the experiment section.

Figure 4: what is the meaning of figure 4b?

Figure 5: Please give a clear explanation of the status of this sample. What kind of test, and how long is the test before lamella is prepared?

Figure 6: Scale bar missing. The area's name and the figure's name are very easy to mislead the reader; please change it to another system like Roman letters.

Figure 7: The EDX scan here lacks explanation. Is it a line scan or a point scan? How do you define the distance between each dot if it is a point scan? Do you have multiple location scans here to ensure the repeatability of data? (error bar needed); Please remake the intensity plotting better since it is hard to read in figure 7b.

Kindly suggest adding other characterization, such as thermogravimetric analysis, to support your conclusion.

Microstructure studies between Ti-TiN-(Ti, Y, Al)N and Ti-TiN-(Ti, Cr, Al)N are missing.

Author Response

Reviewer 1

Comments and Suggestions for Authors

R: After reading this article carefully, I think a major revision is necessary before bringing this paper to the readers.

A: The authors are grateful to the Reviewer for valuable recommendations that improve the quality of the manuscript.

R: First, the title is unclear or cannot precisely cover the topics discussed in this paper. The majority of discussion is regarding the wear-resistant properties of the (Ti,Y,Al)N top layer in this multilayer structure and possible evidence of why this layer can provide such good properties. Please reconsider the name of this article.

A: The name has been changed in accordance with the recommendation of the Reviewer.

R: However, even if the appropriate name has been addressed here, this article still needs significant revision since the evidence to support the conclusion is insufficient, and some vital material characterizations are still missing.

1.Abstract:

I suggest the authors rewrite the abstract. The abstract should bring the motivation of the research and the highlighted part of the conclusion/discovery after revies the experiment part.

A: The abstract has been modified to reflect the recommendations of the Reviewer.

  1. Introduction

R: I suggest the author simplify/rewrite the introduction session and give a more precise/more detailed picture of how it relates to your research motivation/topic. Too many articles have been mentioned; however, not all of them are related, and they are not well organized to fit with the scope of this paper.

A: The Introduction section has been modified and focused on the subject of the study. Some references that are not directly related to the topic of the article were excluded.

R: Please add a clear explanation and background introduction of the Ti/N-based multilayer system in the beginning part of the instruction. The advantages of this multilayer system (address on high wear-resistance part) and how the Y, Cr play inside should be discussed next. The materials from rows 47-116 should be reorganized and simplified after introducing the system.

A: The introduction has been modified in accordance with the recommendations of the Reviewer. Added consideration of the effect of the nanolayer structure on the properties of the coating, as well as the main effect of the introduction of Cr or Y into the composition of coatings based on the (Ti,Al)N system. Since the main attention in the article is paid to the coating based on (Ti,Y,Al)N, the influence of Y on the properties of coatings was considered in detail.

R: Please add the instruction to the deposition method used in this paper and related work.

A: A more detailed description of the deposition technique is provided in the Materials and Methods section.

  1. Materials and Methods

R: The whole experiment setup and analysis are not straightforward for me. Please reorganize and separate this section into two subsections -sample preparation & characterization.

A: Section 3. Materials and Methods is divided into two sub-sections in accordance with the recommendations of the Reviewer.

R: Please keep the model type number in parentheses for all the systems/tools used here.

A: Designations of models and manufacturers are supplemented.

Sample preparation part:

R: 1. Please explain how the sample has been prepared step by step in time and what kind of sample has been prepared/used.

A: The sample preparation sequence is presented in a structured way.

R: 2. Please provide a few schematic drawings here to show the deposition system and sample preparation flow.

A: A detailed circuit diagram and layout of the VIT-2 unit is presented in [46]. This link and related reference have been added.

Sample characterization part:

R: 1. Please explain which kinds of characterization are needed first and why they are needed, then give a description of the tools/settings. For example, the method of transverse section samples (rows 228-234) should be discussed in the sample preparation part.

A: The description has been changed and added in accordance with the recommendations of the Reviewer.

R: 2. Add material to illustrate how the hardness and cutting tests were conducted.

A: The descriptions have been expanded and supplemented in accordance with the recommendations of the Reviewer.

R: 3. reorganize all the paragraphs.

A:  The paragraphs have been reorganized in accordance with the recommendations of the Reviewer.

  1. Results and Discussion

R: Table 2: Please provide the detail of how you obtain the element content in each coating.

A: The elemental composition was determined with the EDX (energy-dispersive X-ray) sys-tem of INCAEnergy (OXFORD Instruments, UK).

R: Figure 1: Please provide an XRD analysis here to support your conclusion.

A: XRD analysis is less accurate than the SAED method. In particular, when using XRD, there are also data on the phases of the substrate (carbide) and the determination of the phase composition is possible with less accuracy and detail. XRD analysis shows only the (Ti,Y,Al)N phase (see below), but SAED clearly shows the presence of two phases (the (Y,Ti,Al)N phase is also observed).

R: Kindly suggest adding the Ti-TiN-( Ti, Y, Al)N with low Y content analysis as evidence to support your conclusion.

A: The (Ti,Y,Al)N coating with low Y content was not investigated in this series of experiments. The study of the effect of the Y content on the properties of the (Ti,Y,Al)N coating is an interesting task and is planned for further work.

R: Figure 2: Please explain the criterion of why the data stopped recording at 7 and 16,16 mins, respectively.

A: The flank wear VBmax = 0.3 mm was assumed as the criterion for the limit wear of the tools. The uncoated tool reached this criterion in 7 minutes and the coated tools in 16 minutes.

R: Figure 3 needs to be reorganized and moved to the experiment section.

A: Figure 3 moved to section 2 as recommended by the Reviewer.

R: Figure 4: what is the meaning of figure 4b?

A: The authors agree with the Reviewer. Figure 4b does not make sense within the framework of this article and, accordingly, has been deleted.

R: Figure 5: Please give a clear explanation of the status of this sample. What kind of test, and how long is the test before lamella is prepared?

A: Description added in accordance with the recommendations of the Reviewer.

R: Figure 6: Scale bar missing.

A: Scale bars added.

R: The area's name and the figure's name are very easy to mislead the reader; please change it to another system like Roman letters.

A: The authors apologize, they apparently did not fully understand the Reviewer's recommendation. Area designations A, B C were used (see Fig. 5), if a smaller area was considered, for example, area C, then it was designated, respectively, as C1 or C2.

R: Figure 7: The EDX scan here lacks explanation.

A: It seems to the authors that the results of the EDX scan are described in sufficient detail in the section preceding Fig. 7. The authors are ready to further expand this description if the Reviewer prompts the direction of additional description.

Is it a line scan or a point scan? How do you define the distance between each dot if it is a point scan?

A: The composition of round regions (dots) located at a distance of 75 - 115 nm from each other was studied. In principle, linear analysis is also an analysis of a sequence of points but built automatically. In this case, the task was to investigate the elemental composition of the main areas (adherent, coating, outer layers of the substrate). In the figure, the areas of study were additionally indicated.

R: Do you have multiple location scans here to ensure the repeatability of data? (error bar needed);

A: Since the method provides sufficient measurement accuracy for the purposes of the study, the measurements were not repeated. Accordingly, there are no error bars. The most interesting areas (areas 6 and 7 in Figure 7b) were further explored.

R: Please remake the intensity plotting better since it is hard to read in figure 7b.

A: The figure has been modified in accordance with the recommendations of the Reviewer.

R: Kindly suggest adding other characterization, such as thermogravimetric analysis, to support your conclusion.

Unfortunately, thermogravimetric analysis was not performed in this series of experiments. This analysis is of undoubted interest. However, it should be noted that the conditions in the cutting zone have a number of differences from "pure" thermogravimetry. In particular, diffusion processes and the effect of the chip flow on the formed oxides take place.

R: Microstructure studies between Ti-TiN-(Ti, Y, Al)N and Ti-TiN-(Ti, Cr, Al)N are missing.

A: Since the objective of the study was precisely the Ti-TiN-(Ti, Y, Al)N coating, and the Ti-TiN-(Ti, Cr, Al)N coating was considered only as an object of comparison in cutting tests, detailed metallographic studies of the Ti- TiN-(Ti, Cr, Al)N were not carried out. The authors agree that such comparative studies could be of undoubted interest.

Reviewer 2 Report

In the current work, the authors investigated microstructural peculiarities as well as the wear behavior and mechanisms of Y-rich (Ti,Al,Y)N coating (Y content of about 40 at.% on the metal sublattice). The cutting tests revealed that this coating performed nearly as well as the benchmark (Ti,Cr,Al)N coating, which is widely used in the manufacturing of metal-cutting tools, and significantly outperformed the uncoated tool. Although the results might be of considerable interest for the community, the manuscript requires a substantial revision prior to publication.

1. The introduction and methodology sections make up more than half of the manuscript, whereas the most important section (results and discussion) is disproportionally short – just about one third. Not only the introduction section is unnecessarily long but is incoherent, too. Consider shortening the introduction sections and simultaneously expanding on the results and discussion.

2. About one quarter of the references are self-citations. Some of them seem to be out of place, e.g., line 150 reads “… and some amount of Ti and Al will be dissolved in c-YN [52, 53]”. Refs. 52 and 53 are self-citations, the articles are entitled “Investigation of the tribological properties of Ti-TiN-(Ti,Al,Nb,Zr)N composite coating and its efficiency in increasing wear resistance of metal cutting tools” and “Investigation of wear mechanisms of multilayer nanostructured wear-resistant coatings during turning of steel. Part 2: Diffusion, oxidation processes and cracking in Ti-TiN-(Ti,Cr,Mo,Al)N coating”, respectively. So, how are these studies related to the solubility of Ti and Al in c-YN?

3. The manuscript would also need some careful proofreading and further elucidation. Just to name a few examples:

“the protective functions of the layers with the dominance of (Ti,Al)N, which protect the underlying layers of (Y,Ti,Al)N from early oxidative damage” (lines 32-33 and 316-317) – The overall content of Al is 7.69 at.%, how effective is such (Ti,Al)N in increasing oxidation resistance? 

"A possible useful property of YN and Y2O3 is their relatively high fracture toughness" (line 124-125) – Fracture toughness was not under investigation in Refs. [32,33,37], was it? Given that Y2O3 is soft and forms on the surface (e.g. poor resistance against abrasive wear), why is its fracture toughness considered useful?

“Combined with high fracture toughness (for example, the layers based on YN)” (line 129) – How high is fracture toughness of YN? As this was the motivation for the study, indicate the numbers and put this into perspective.

"the very nature of the physical vapor deposition (PVD) process suggests that these phases will not be pure, but some amount of Y will be dissolved in c-(Ti,Al)N, and some amount of Ti and Al will be dissolved in c-YN" (lines 148-150) – It's not in PVD's nature at all. In fact, even atomically sharp interfaces can be produced in a PVD process.

All in all, regretfully, I cannot recommend the manuscript for publication in its current form.

Author Response

Reviewer 2

Comments and Suggestions for Authors

In the current work, the authors investigated microstructural peculiarities as well as the wear behavior and mechanisms of Y-rich (Ti,Al,Y)N coating (Y content of about 40 at.% on the metal sublattice). The cutting tests revealed that this coating performed nearly as well as the benchmark (Ti,Cr,Al)N coating, which is widely used in the manufacturing of metal-cutting tools, and significantly outperformed the uncoated tool. Although the results might be of considerable interest for the community, the manuscript requires a substantial revision prior to publication.

  1. The introduction and methodology sections make up more than half of the manuscript, whereas the most important section (results and discussion) is disproportionally short – just about one third. Not only the introduction section is unnecessarily long but is incoherent, too. Consider shortening the introduction sections and simultaneously expanding on the results and discussion.

R: 2. About one quarter of the references are self-citations. Some of them seem to be out of place, e.g., line 150 reads “… and some amount of Ti and Al will be dissolved in c-YN [52, 53]”. Refs. 52 and 53 are self-citations, the articles are entitled “Investigation of the tribological properties of Ti-TiN-(Ti,Al,Nb,Zr)N composite coating and its efficiency in increasing wear resistance of metal cutting tools” and “Investigation of wear mechanisms of multilayer nanostructured wear-resistant coatings during turning of steel. Part 2: Diffusion, oxidation processes and cracking in Ti-TiN-(Ti,Cr,Mo,Al)N coating”, respectively. So, how are these studies related to the solubility of Ti and Al in c-YN?

A: This fragment has been removed, as well as the corresponding references. In the presented articles, the gradient nature of coating nanolayers was considered, but the coatings did not include yttrium.

  1. The manuscript would also need some careful proofreading and further elucidation. Just to name a few examples:

R: “the protective functions of the layers with the dominance of (Ti,Al)N, which protect the underlying layers of (Y,Ti,Al)N from early oxidative damage” (lines 32-33 and 316-317) – The overall content of Al is 7.69 at.%, how effective is such (Ti,Al)N in increasing oxidation resistance?

A: I think that even TiN without aluminum can effectively isolate YN from water and oxygen. TiN unlike YN has good oxidation resistance and does not react with water. That is, the YN layers under the TiN layer will not react with water/oxygen until the TiN layer breaks down.

R: "A possible useful property of YN and Y2O3 is their relatively high fracture toughness" (line 124-125) – Fracture toughness was not under investigation in Refs. [32,33,37], was it? Given that Y2O3 is soft and forms on the surface (e.g. poor resistance against abrasive wear), why is its fracture toughness considered useful?

A: Unfortunately, not many articles have been found on the fracture toughness of yttrium oxide/nitride. Here are a couple of related articles:

  1. Am. Ceram. Soc. 1989, 72(8), 1562-1563.

Surf. Coat. Technol. 2017, 321, 57-63.

It turns out that the fracture toughness is at least noticeably higher than that of TiN or TiAlN. Given the layered structure of the coating, the alternation of harder TiAlN layers and softer YN layers can inhibit the development of cracks. composite effect. Of course, this requires more careful study.

R: “Combined with high fracture toughness (for example, the layers based on YN)” (line 129) – How high is fracture toughness of YN? As this was the motivation for the study, indicate the numbers and put this into perspective.

A: Fracture toughness increases significantly (from 0.1 to 1.0 MPa×m1/2 with increasing Y content from 0 to 20 at%) with an increase in the yttrium content in the TiN-based coating [39]. Clarification added.

R: "the very nature of the physical vapor deposition (PVD) process suggests that these phases will not be pure, but some amount of Y will be dissolved in c-(Ti,Al)N, and some amount of Ti and Al will be dissolved in c-YN" (lines 148-150) – It's not in PVD's nature at all. In fact, even atomically sharp interfaces can be produced in a PVD process.

A: The authors removed this maxim. In this case, the authors, as a rule, encounter precisely the gradient distribution of the content of elements in nanolayers. This deserves a separate study and is not fundamental for this article.

R: All in all, regretfully, I cannot recommend the manuscript for publication in its current form.

A: The authors tried to take into account all the recommendations of the Reviewer and improve the quality of the manuscript.

Reviewer 3 Report

The authors studied the tribological behaviors of the Ti-TiN-(Ti,Y,Al)N multilayer composite coating with the high content of yttrium (Y) in the wear-resistant layer. The related mechanisms were also revealed. This manuscript is of high quality. Hence, I recommend this manuscript to be accepted by coatings.

Author Response

Reviewer 3

Comments and Suggestions for Authors

R: The authors studied the tribological behaviors of the Ti-TiN-(Ti,Y,Al)N multilayer composite coating with the high content of yttrium (Y) in the wear-resistant layer. The related mechanisms were also revealed. This manuscript is of high quality. Hence, I recommend this manuscript to be accepted by coatings.

A: The authors are grateful to the Reviewer for the high appreciation of their work.

Reviewer 4 Report

The manuscript presents original and interesting results of microstructure and wear resistance of Ti-TiN-(Ti,Y,Al)N multilayer coating. However, the structure of the article could be improved before the paper acceptance for publication. Introduction section seems to be too long (as compared with other sections of the article) and written in quite chaotical way. So, it should be rewritten in the way that a single paragraph covers a literature data important from the point of view of this article, being a reference for the presented study. In Results and Discussion section the Authors failed to provide general (cross-sectional) overview (SEM images) of the as-deposited coating.  

Author Response

Reviewer 4

Comments and Suggestions for Authors

R: The manuscript presents original and interesting results of microstructure and wear resistance of Ti-TiN-(Ti,Y,Al)N multilayer coating. However, the structure of the article could be improved before the paper acceptance for publication.

A: The authors are grateful to the Reviewer for their help in improving the quality of the manuscript.

R: Introduction section seems to be too long (as compared with other sections of the article) and written in quite chaotical way. So, it should be rewritten in the way that a single paragraph covers a literature data important from the point of view of this article, being a reference for the presented study.

A: The introduction has been modified and substantially revised in accordance with the recommendations of the Reviewers.

R: In Results and Discussion section the Authors failed to provide general (cross-sectional) overview (SEM images) of the as-deposited coating.  

A: Cross sections (SEM) of compared coatings are added (Fig. 2 a,b)

Reviewer 5 Report

I would like to congratulate the authors for the work done.

The title of the manuscript is appropriate and concise and the work fits within the scope of the journal. The language of the paper is clearly written. The manuscript contains substantially new and interesting information that is of sufficient importance to justify publication.

The Abstract reflects the paper content and summarizes the problem, the method, the results, and the conclusions.

The purpose of the study is clearly outlined, and the results of previous work discussed in the introduction (chapter 1).

In Materials and Methods, the author explains precisely how the data were collected, however, information is lacking so that the experiment can be reproduced. It would be important for the authors to add: the number of samples used, the size of the samples, the purity of the gas used, and the deposition temperature.

In Results and Discussion: The Results are duly explained, and the authors show how the work resulted in a breakthrough in the study of coatings with high yttrium content. The discussion is supported by the results and makes scientific sense; however, the authors could have referred to more comparisons between current results with other authors, to improve their work.

The authors should add a reference in line 197:  "At the elastic modulus of 356 ± 24 GPa, the hardness of the (Ti,Y,Al)N coating is considerably high (HV 2758 ± 78). The reference coating of (Ti,Cr,Al)N has the hardness of HV 3182 ± 53 and the elastic modulus of 438 ± 32 GPa. "

Authors should refer in figures 5, 6, and 7 the equipment for capturing these images (TEM), following the same logic as in the previous images.

The conclusions are sound and justifiable based on the results and the discussion. The authors left no recommendations for future work, which is a shame. Given the importance of the research carried out by the authors, it would be very important for the authors to recommend future works, so that other members of the scientific community could develop other works in the future, based on this research. The authors also did not point out any limitations that this research had during its development.

Author Response

Reviewer 5

Comments and Suggestions for Authors

R: I would like to congratulate the authors for the work done.

A: The authors are grateful to the Reviewer for the high appreciation of their work.

The title of the manuscript is appropriate and concise and the work fits within the scope of the journal. The language of the paper is clearly written. The manuscript contains substantially new and interesting information that is of sufficient importance to justify publication.

The Abstract reflects the paper content and summarizes the problem, the method, the results, and the conclusions.

The purpose of the study is clearly outlined, and the results of previous work discussed in the introduction (chapter 1).

A: The authors are grateful to the Reviewer for the high appreciation of their work.

R: In Materials and Methods, the author explains precisely how the data were collected, however, information is lacking so that the experiment can be reproduced. It would be important for the authors to add: the number of samples used, the size of the samples, the purity of the gas used, and the deposition temperature.

A: Information added:

Chemically pure nitrogen of high purity (grade 6.0) was used. Volume fraction of nitrogen, % not less than 99.99990.

The substrate used was carbide inserts SNUN ISO 1832:2012, 12.00 × 12.00 × 4.75 mm (WC+15% TiC+6% Co) (KZTS, Kirovograd, Russia). Each type of coating was depos-ited on 8 samples.

The surface temperature of the substrate during the deposition of the coating was 650 – 700 °C.

R: In Results and Discussion: The Results are duly explained, and the authors show how the work resulted in a breakthrough in the study of coatings with high yttrium content. The discussion is supported by the results and makes scientific sense; however, the authors could have referred to more comparisons between current results with other authors, to improve their work.

A: The article has been substantially modified and supplemented in accordance with the recommendations of the Reviewers.

R: The authors should add a reference in line 197:  "At the elastic modulus of 356 ± 24 GPa, the hardness of the (Ti,Y,Al)N coating is considerably high (HV 2758 ± 78). The reference coating of (Ti,Cr,Al)N has the hardness of HV 3182 ± 53 and the elastic modulus of 438 ± 32 GPa. "

A: According to the authors, there is no need for a reference in this place. "Reference" coverage, meaning "comparison object".

R: Authors should refer in figures 5, 6, and 7 the equipment for capturing these images (TEM), following the same logic as in the previous images.

A: An imaging method (TEM) designator has been added.

R: The conclusions are sound and justifiable based on the results and the discussion. The authors left no recommendations for future work, which is a shame. Given the importance of the research carried out by the authors, it would be very important for the authors to recommend future works, so that other members of the scientific community could develop other works in the future, based on this research. The authors also did not point out any limitations that this research had during its development.

A: As a further line of research, the authors plan to investigate the effect of yttrium content on coating properties, also considering coatings with lower yttrium content. This maxim has been added to the Conclusion.

Round 2

Reviewer 1 Report

There are still some sections that need to improve here.

1. Row 27-36. The Abstract should focus on the highlighted conclusion of the research. I am unsure why the author copied this from rows 357-364 since there is just a hypothesis and no clear evidence yet. 

2. The introduction section is still too long. I understand the author wants to explain how the Y/Cr improves the tool life from rows 56-140. However, this part is still too long and not well-organized. Please give a summative sentence at the very beginning of each paragraph so each paragraph can be used to explain the evidence one by one. Please make sure all the details you want to mention in the introduction part is “tightly” related to the topic. I did not see the logic in this section (how it related to your research motivation) but just many literature reviews.

3. Please clarify the deposition flow of each layer on the sample since multiple targets are used here. (Ti layer/TiN layer/Ti Y Al Layer )

4. I am unsure why the author abruptly brings up the plan for future work (rows 365-366). The author should address the weak parts of the current work and motivate the reader regarding the possible way to improve the coating from different perspectives. Please try to combine it with the discussion section.

Author Response

Reviewer 1:  Comments and Suggestions for Authors

R: There are still some sections that need to improve here.

A: The authors are grateful to the Reviewer for his careful and profound work and for his help in improving the quality of the manuscript.

R: 1. Row 27-36. The Abstract should focus on the highlighted conclusion of the research. I am unsure why the author copied this from rows 357-364 since there is just a hypothesis and no clear evidence yet.

A: The abstract has been changed.

R: 2. The introduction section is still too long. I understand the author wants to explain how the Y/Cr improves the tool life from rows 56-140. However, this part is still too long and not well-organized. Please give a summative sentence at the very beginning of each paragraph so each paragraph can be used to explain the evidence one by one. Please make sure all the details you want to mention in the introduction part is “tightly” related to the topic. I did not see the logic in this section (how it related to your research motivation) but just many literature reviews.

A: The introduction has been significantly changed and supplemented. The authors hope that in a modified form, the motivation, goals and objectives of the work are more justified and understandable.

R: 3. Please clarify the deposition flow of each layer on the sample since multiple targets are used here. (Ti layer/TiN layer/Ti Y Al Layer )

A: In the deposition of adhesive and transition layers, only the Ti cathode was used; deposition took place in an argon and nitrogen atmosphere, respectively. When depositing (Ti,Y,Al)N wear-resistant layer, three cathodes were used: Ti, Y, and Al. Description added to the manuscript.

R: 4. I am unsure why the author abruptly brings up the plan for future work (rows 365-366). The author should address the weak parts of the current work and motivate the reader regarding the possible way to improve the coating from different perspectives. Please try to combine it with the discussion section.

A: The relevant paragraph has been expanded and moved to the Discussion section.

Reviewer 2 Report

I. Unnecessary long introduction

R2: The introduction and methodology sections still make up more than half of the manuscript, whereas the most important section (results and discussion) is disproportionally short – less than one third. The introduction section is still unnecessarily long and, sadly, unreadable as there is no structure at all. Again, please consider shortening the introduction sections and simultaneously expanding on the results and discussion.

II. Self-references.

R2: The main point of concern was the number of self- references, though. About one quarter of the references are self-citations – a good rule of thumb is that self-references should lie below 10 % of the total amount of citations.

About one quarter of the references are self-citations. Some of them seem to be out of place, e.g., line 150 reads “… and some amount of Ti and Al will be dissolved in c-YN [52, 53]”. Refs. 52 and 53 are self-citations, the articles are entitled “Investigation of the tribological properties of Ti-TiN-(Ti,Al,Nb,Zr)N composite coating and its efficiency in increasing wear resistance of metal cutting tools” and “Investigation of wear mechanisms of multilayer nanostructured wear-resistant coatings during turning of steel. Part 2: Diffusion, oxidation processes and cracking in Ti-TiN-(Ti,Cr,Mo,Al)N coating”, respectively. So, how are these studies related to the solubility of Ti and Al in c-YN?

A: This fragment has been removed, as well as the corresponding references. In the presented articles, the gradient nature of coating nanolayers was considered, but the coatings did not include yttrium.

III. Fracture toughness.

R2: It appears to be prime motivation behind the study (lines 122-125: "Of particular interest could be an investigation focused on the properties of a coating in which the layers with high hardness and wear resistance (for example, the layers based on the (Ti,Al)N system) would be alternated with the layers with good barrier properties combined with high fracture toughness (for example, the layers based on YN)"). However, there is no evidence supporting the statement that YN has a high fracture toughness. In fact, in Ref. 39, only apparent fracture toughness is reported which strongly correlates with the residual stresses in the coatings. Besides, fracture toughness of TiN is considerably higher than the apparent fracture toughness reported in Ref. 39 (in which it is strongly affected by the tensile residual stresses).

R: "A possible useful property of YN and Y2O3 is their relatively high fracture toughness" (line 124-125) – Fracture toughness was not under investigation in Refs. [32,33,37], was it? Given that Y2O3 is soft and forms on the surface (e.g. poor resistance against abrasive wear), why is its fracture toughness considered useful?

A: Unfortunately, not many articles have been found on the fracture toughness of yttrium oxide/nitride. Here are a couple of related articles:

Am. Ceram. Soc. 1989, 72(8), 1562-1563.

Surf. Coat. Technol. 2017, 321, 57-63.

It turns out that the fracture toughness is at least noticeably higher than that of TiN or TiAlN. Given the layered structure of the coating, the alternation of harder TiAlN layers and softer YN layers can inhibit the development of cracks. composite effect. Of course, this requires more careful study.

R: “Combined with high fracture toughness (for example, the layers based on YN)” (line 129) – How high is fracture toughness of YN? As this was the motivation for the study, indicate the numbers and put this into perspective.

A: Fracture toughness increases significantly (from 0.1 to 1.0 MPa×m1/2 with increasing Y content from 0 to 20 at%) with an increase in the yttrium content in the TiN-based coating [39]. Clarification added.

Author Response

Reviewer 2: Comments and Suggestions for Authors

A: The authors are grateful to the Reviewer for his careful and profound work and for his help in improving the quality of the manuscript.

  1. Unnecessary long introduction

R2: The introduction and methodology sections still make up more than half of the manuscript, whereas the most important section (results and discussion) is disproportionally short – less than one third. The introduction section is still unnecessarily long and, sadly, unreadable as there is no structure at all. Again, please consider shortening the introduction sections and simultaneously expanding on the results and discussion.

A→R2: Now the structure of the article (without references and title page) is as follows:

Introduction - 2 pages (was noticeably shortened and restructured)

Materials and Methods - 2 pages (was significantly expanded at the request of the Reviewers)

Results and Discussion + Conclusions - 6 pages

  1. Self-references.

R2: The main point of concern was the number of self- references, though. About one quarter of the references are self-citations – a good rule of thumb is that self-references should lie below 10 % of the total amount of citations.

A→R2: In accordance with the recommendations of the Reviewer, the number of self-citations has been reduced.

About one quarter of the references are self-citations. Some of them seem to be out of place, e.g., line 150 reads “… and some amount of Ti and Al will be dissolved in c-YN [52, 53]”. Refs. 52 and 53 are self-citations, the articles are entitled “Investigation of the tribological properties of Ti-TiN-(Ti,Al,Nb,Zr)N composite coating and its efficiency in increasing wear resistance of metal cutting tools” and “Investigation of wear mechanisms of multilayer nanostructured wear-resistant coatings during turning of steel. Part 2: Diffusion, oxidation processes and cracking in Ti-TiN-(Ti,Cr,Mo,Al)N coating”, respectively. So, how are these studies related to the solubility of Ti and Al in c-YN?

A: This fragment has been removed, as well as the corresponding references. In the presented articles, the gradient nature of coating nanolayers was considered, but the coatings did not include yttrium.

III. Fracture toughness.

R2: It appears to be prime motivation behind the study (lines 122-125: "Of particular interest could be an investigation focused on the properties of a coating in which the layers with high hardness and wear resistance (for example, the layers based on the (Ti,Al)N system) would be alternated with the layers with good barrier properties combined with high fracture toughness (for example, the layers based on YN)"). However, there is no evidence supporting the statement that YN has a high fracture toughness. In fact, in Ref. 39, only apparent fracture toughness is reported which strongly correlates with the residual stresses in the coatings. Besides, fracture toughness of TiN is considerably higher than the apparent fracture toughness reported in Ref. 39 (in which it is strongly affected by the tensile residual stresses).

A→R2: The introduction has been significantly changed and supplemented. The authors hope that in a modified form, the motivation, goals and objectives of the work are more justified and understandable.

R: "A possible useful property of YN and Y2O3 is their relatively high fracture toughness" (line 124-125) – Fracture toughness was not under investigation in Refs. [32,33,37], was it? Given that Y2O3 is soft and forms on the surface (e.g. poor resistance against abrasive wear), why is its fracture toughness considered useful?

A: Unfortunately, not many articles have been found on the fracture toughness of yttrium oxide/nitride. Here are a couple of related articles:

Am. Ceram. Soc. 1989, 72(8), 1562-1563.

Surf. Coat. Technol. 2017, 321, 57-63.

It turns out that the fracture toughness is at least noticeably higher than that of TiN or TiAlN. Given the layered structure of the coating, the alternation of harder TiAlN layers and softer YN layers can inhibit the development of cracks. composite effect. Of course, this requires more careful study.

R: “Combined with high fracture toughness (for example, the layers based on YN)” (line 129) – How high is fracture toughness of YN? As this was the motivation for the study, indicate the numbers and put this into perspective.

A: Fracture toughness increases significantly (from 0.1 to 1.0 MPa×m1/2 with increasing Y content from 0 to 20 at%) with an increase in the yttrium content in the TiN-based coating [39]. Clarification added.

Reviewer 4 Report

The paper was improved and may be accepted in the present form.

Author Response

A: The authors are grateful to the Reviewer for his careful and profound work and for his help in improving the quality of the manuscript.